# Genome-Wide Identification of Sucrose Transporter Genes and Functional Analysis of *RsSUC1b* in Radish (*Raphanus sativus* L.)

Xiaofeng Zhu [1,†], Xiaoli Zhang [1,†], Yang Cao [1,†], Ruixian Xin [1], Yinbo Ma [2], Lun Wang [2], Liang Xu [1], Yan Wang [1], Rui Liu [1] and Liwang Liu [1,2,*]

1   National Key Laboratory of Crop Genetics and Germplasm Enhancement, Key Laboratory of Horticultural Crop Biology and Genetic Improvement (East China), Ministry of Agriculture and Rural Affairs of P.R.C., College of Horticulture, Nanjing Agricultural University, Nanjing 210095, China
2   College of Horticulture and Landscape Architecture, Yangzhou University, Yangzhou 225009, China
*   Correspondence: nauliulw@njau.edu.cn
†   These authors contributed equally to this work.

**Abstract:** In most higher plants, sucrose is the significant form of carbohydrate for long-distance transportation. Sucrose transporters/sucrose carriers (SUTs/SUCs) are involved in the loading and unloading of sucrose in phloem and play an important role in the growth and development of plants. In this study, 12 *RsSUC* genes were first identified from the radish genome, and their phylogenetic relationships, gene structure, and conserved motifs were further analyzed. RT-qPCR results indicated that *RsSUC* genes exhibited various expression patterns in different tissues and development stages of the radish. Overexpression of *RsSUC1b* in *Arabidopsis* significantly improved the uptake efficiency of exogenous sucrose, and promoted leaves and lateral root growth. In addition, the transgenic plants flowered significantly earlier than wild-type (WT) plants, and the soluble sugar contents (SSCs) including sucrose, glucose, and fructose in the mature leaves and pods were increased. It could be inferred that RsSUC1b is a plasma membrane sucrose transporter and plays a vital role in sucrose transportation and sugar accumulation during plant growth and development. These findings provided novel insights into the biological function of *RsSUC* genes and facilitate dissecting the molecular mechanism underlying sugar transport during radish development.

**Keywords:** radish; *SUC* gene family; expression analysis; *RsSUC1b*; functional analysis





## 1. Introduction

The radish (*Raphanus sativus* L.) is widely cultivated worldwide, and its thickened taproot is not only a product organ, but also an important sink organ [1]. Radish taproot is rich in nutrients, including protein, carbohydrates, minerals, vitamins, lipids, and amino acids. Carbohydrates mainly include glucose, sucrose, and fructose [2]. Sucrose is the main product of photosynthesis in higher plants and the main form of assimilated transport in phloem [3]. Sucrose export directly affects the transportation and distribution of photosynthate, which plays a critical role in the formation and development of radish taproot [4]. Therefore, understanding the molecular mechanism of sucrose transportation in radishes is of great significance to improve the yield and quality of radishes.

Sucrose is a kind of soluble sugar, which is the main photosynthate in the leaves of higher plants. It can maintain the metabolism of source organs; most products are loaded into phloem and distributed to plant sink organs, such as the stem, seed, and taproot [5,6]. The transport of sucrose involves symplast and apoplast pathways in phloem [7]. In the apoplast pathway, sucrose loading or unloading requires the participation of sucrose transporters/sucrose carriers (SUTs/SUCs) [8,9].

SUCs are a class of carrier proteins that can mediate sucrose transport, and widely exist in tissues and cells of higher plants [10]. SUCs were members of a subfamily of the glycoside-pentoside-hexuronide (GPH) cation family in the major facilitator superfamily

(MFS) [11] and played a critical role in the transportation of soluble sucrose. SUC gene was first isolated from spinach, and it was found that the gene had sucrose transport ability through the transformation of INV-deficient yeast experiment [12]. Until now, lots of SUC genes were successfully isolated from various plants including sorghum [13], oilseed rape [14], pomegranate [15], and blackberry [16]. SUCs in dicotyledons could be classified into three subgroups, SUT1, SUT2, and SUT4 [17]. The influence of SUT1 members on plant growth and development has been verified in a variety of plants by gain or loss of function analysis. The *Arabidopsis* sucrose transporter *AtSUC1* was found to function in sucrose transport during development of pollen and roots [18,19]. Overexpressing of the spinach *SoSUT1* gene in potatoes could promote the export of sucrose in leaf and significantly increase the content of soluble sugar in tubers [20]. Additionally, inhibiting the expression of the maize *ZmSUT1* gene resulted in a large amount of soluble sugar accumulation in leaves, and the leaves withered and became senescent rapidly, and plant flowering was delayed [21]. These results demonstrated the important role of *SUC* genes in regulating sucrose transport during plant growth and development.

In this study, to further investigate the function of *RsSUC* genes involved in regulating sucrose transport in radish, SUC family members were first identified from the radish genome, and the sequence characteristics, gene structure, conserved motifs, transmembrane regions, and phylogenetic relationships were analyzed. In addition, the expression profiles of *RsSUC* genes were explored in the leaves and taproots at different developmental stages. Additionally, the function of *RsSUC1b* was further investigated by overexpression in *Arabidopsis*. These results provided novel insight into the function of *SUC* genes in sucrose transport during radish taproot formation.

## 2. Materials and Methods

### 2.1. Plant Materials

An advanced inbred line, NAU-ZYQ, was sown in the NAU (Baima) experimental station. Tissue samples of leaf and root were harvested every 20 days after sowing (20 DAS) until harvest (100 DAS). The phenotypes during the development stages of the radish were shown in Figure S1. The growth cycle of radishes is about three months. According to the phenotypic changes in each stage, the development stages of radishes could be divided into cortex split stage (20–40 DAS), early stage of taproot thickening (40–60 DAS), middle stage of taproot thickening (60–80 DAS), and late stages of taproot thickening (80–100 DAS). An approximately 0.3 g sample was rapidly frozen in liquid nitrogen and stored at −80 °C for gene expression analysis, and the remaining sample was dried at 80 °C for soluble sugar content determination.

Tobacco (*Nicotiana benthamiana*) was utilized for the subcellular localization analysis. *A. thaliana* (ecotype: Col-0) was utilized for genetic transformation.

### 2.2. Identification of SUC Gene Family in Radish

The SUC protein sequences containing conserved domain (PF13347) in the radish genome (WK10039, http://radish-genome.org/) (accessed on 5 July 2021) were searched by Hmmer3.0 software [22]. Subsequently, NCBI (https://www.ncbi.nlm.nih.gov/Structure/bwrpsb/bwrpsb.cgi) (accessed on 12 July 2021), SMART (http://smart.embl-heidelberg.de/) (accessed on 12 July 2021), and Pfam (http://Pfam.sanger.ac.uk/) (accessed on 12 July 2021) were employed to further verify the conserved domain of candidate SUC proteins.

### 2.3. Sequence Characteristic and Phylogenetic Analysis

The ExPASy ProtParam (http://web.expasy.org/protparam/) (accessed on 20 August 2021) was performed to predict relative molecular weight, theoretical pI, instability index, aliphatic index, and grand average of hydropathicity of RsSUC proteins [23]. Moreover, the transmembrane regions were analyzed by Hidden Markov Models Server v.2.0 (http://www.cbs.dtu.dk/services/TMHMM/) (accessed on 21 August 2021) [24]. The *Arabidopsis* SUC protein sequences are from the TAIR database (http://www. arabidopsis.org/)

(accessed on 3 September 2021) (Table S2). The SUC protein sequences of *Brassica oleracea*, *Oryza sativa*, and *Zea mays* were obtained from NCBI (https://www.ncbi.nlm.nih.gov/) (accessed on 9 September 2021) according to the reported *SUC* genes accession numbers (Table S2). The phylogenetic tree including five species was constructed by MEGA6.0 with the neighbor-joining (NJ) and bootstrap value set to 1000 replicates [25]. Additionally, the gene structure was analyzed by TBtools software and conservative motifs were analyzed by MEME (http://meme.sdsc.edu/meme/) (accessed on 15 September 2021) [26].

### 2.4. RNA Extraction and RT-qPCR Analysis

Total RNA was extracted by using the TRIzol reagent RNA simple total RNA kit (Tiangen, Beijing, China). and reverse transcribed into cDNA by using the PrimeScript™ RT reagent kit (Takara, Dalian, China) according to the instructions. The cDNA was used for RT-qPCR to analyze the expression of *RsSUC* genes. RT-qPCR analysis was performed on LightCycler® 480 System (Roche, Mannheim, Germany). *RsActin* was an internal reference gene, and the $2^{-\Delta\Delta CT}$ formula was used to calculate the relative expression level [27,28]. The RT-qPCR primers were shown in Table S3.

### 2.5. Subcellular Localization of RsSUC1b

The CDS of *RsSUC1b* was isolated and deposited into GenBank (Acc. No.: OP653780). The gene construct *35::RsSUC1b-GFP* was transformed into *Agrobacterium tumefaciens* GV3101 strain and was injected into *N. benthamiana* leaves with *35S*::GFP and *35S::AUX1-RFP*, which were used as the negative control and nuclear marker, respectively [29–31]. The subcellular localization of *35S::RsSUC1b-GFP* in the *N. benthamiana* leaf cell was observed under a laser confocal microscope (LSM 800, Zeiss, Jena, Germany). The primer sequences for *35::RsSUC1b-GFP* construction were listed in Table S4.

### 2.6. Construction of Expression Vector and Genetic Transformation

The fusion plasmids *35S::RsSUC1b* was transferred into *Arabidopsis* by using *A. tumefaciens*-mediated transformation with the floral dip method [32]. The transformed seeds were sown on MS solid medium supplemented with 36 mg·L$^{-1}$ hygromycin. Hygromycin-resistant seedlings (T$_1$) were grown to maturity for seed collection. T$_2$ seeds were obtained from selfing T$_1$ plants, and a portion of the T$_2$ seeds was germinated on MS solid medium supplemented with 36 mg·L$^{-1}$ hygromycin. T$_3$ seeds were harvested from T$_2$ plants and used in this study.

### 2.7. Determination of Soluble Sugar Contents

After adding 0.05 g dried samples and 5 mL extracting solution, ultrasonic extraction was performed for 20 min. After an 80 °C water bath for 30 min and 5000 *g* spinning for 15 min, the supernatant was filtered through a 0.45-μm water system filter membrane. The content of soluble sugar was determined by UPLC [3]. Total sugar contents were calculated as the sum of sucrose, glucose, and fructose content.

### 2.8. Statistical Analysis

Data processing was performed by using Microsoft Excel. A Student's *t*-test was used to determine statistical significance between two groups. One-way analysis of variance with least significant difference (LSD) tests were used to determine statistical significance among multiple range tests. Bar graphs were created by using Graphpad Prism software.

## 3. Results

### 3.1. Identification of the Radish SUC Gene Family

A total of 12 *RsSUC* genes on chromosome R1, R2, R4, R5, R6, R7, R9, and RUS were identified from the radish genome (WK10039) according to the combination of Blastp and Hmmer3.0. These *RsSUCs* were named as *RsSUC1a* to *RsSUC4b* (Table S1). The physical and chemical properties of the RsSUC proteins were analyzed. The sizes of

RsSUC proteins ranged from 392 to 540 amino acid (aa) with a molecular weight range from 41.94 to 62.41 kDa, and the theoretical pI ranged from 5.49 to 9.40. The instability index ranged from 29.04 to 40.89, and the aliphatic index varied from 96.64 to 111.00. In addition, according to the grand average of hydropathicity range (0.342~0.545), which suggests that all RsSUC proteins were hydrophobic proteins. TMHMM was used to predict the transmembrane regions of RsSUC proteins. The number of transmembrane regions of RsSUC proteins ranged from 7 to 12, RsSUC2a, RsSUC2b, RsSUC4a, and RsSUC4b contained typical 12 transmembrane regions, whereas RsSUC2c and RsSUC3b had fewer transmembrane regions (Table S1).

### 3.2. Phylogenetic Analysis of RsSUC Proteins

An unrooted phylogenetic tree from radish, *Arabidopsis*, *B. oleracea*, *O. sativa*, and *Z. mays* was created to understand the SUC gene evolutionary relationships (Figure 1). It was indicated that the SUC proteins from monocotyledon and dicotyledon species can be classified into five subgroups including SUT1, SUT2, SUT3, SUT4, and SUT5. The largest was the SUT1 subgroup and was unique to dicotyledons, which consisted of seven members of radish, seven members of *Arabidopsis*, and eight members of *B. oleracea*. SUT3 and SUT5 were specific subgroups of monocotyledons, which consisted of three members of *O. sativa* and four members of *Z. mays*. SUT2 and SUT4 contained both dicotyledon and monocotyledon, which contained five members of radish, two members of *Arabidopsis*, four members of *B. oleracea*, two members of *O. sativa*, and two members of *Z. mays*. These RsSUC proteins were clustered closely with the dicotyledon and were significantly distant from the monocotyledon.

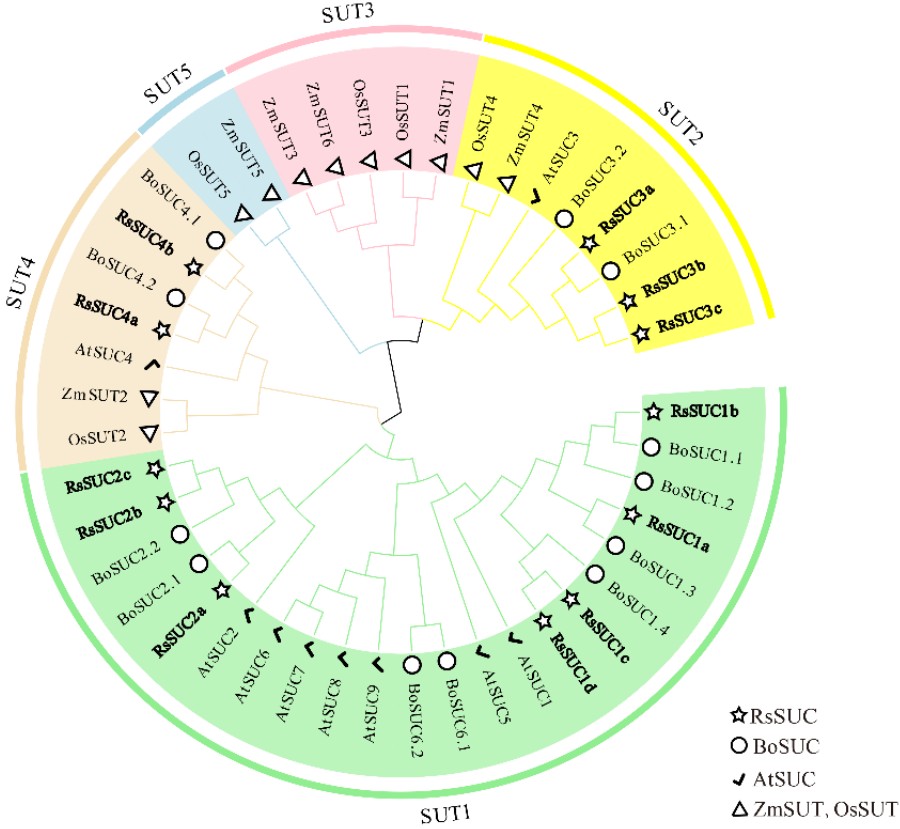

**Figure 1.** Phylogenetic analysis of SUC genes in radish, *Arabidopsis*, *B. oleracea*, *O. sativa*, and *Z. mays*. SUC proteins were divided into five subgroups (SUT1, SUT2, SUT3, SUT4, and SUT5), and each color group represented a subgroup. Stars, circles, checkmarks, and triangles represent SUC proteins in radish, *B. oleracea*, *Arabidopsis*, *O. sativa*, and *Z. mays*, respectively.

### 3.3. Gene Structure and Conserved Motifs Analysis of RsSUCs

RsSUC proteins were classified into three subgroups, SUT1, SUT2, and SUT4 (Figure 2A). The 15 conserved motifs in RsSUC proteins were predicted (Figure 2B). Most RsSUC members of the same subgroup have similar motif compositions. For instance, motif 8 and motif 14 were distributed in SUT1, motif 15 was distributed in SUT2 and SUT4, and motif 12 was only detected in SUT2. These specific motifs might imply diverse functions of the SUC family in radish. The exon-intron structures were further analyzed to understand the structural characteristics of *RsSUC* genes (Figure 2C). The structures of genes in the same subgroup were similar, whereas the number of the exon and intron was different. The number of *RsSUC* exons ranged from 1 to 14. The *RsSUC* genes in SUT1 contained 1 to 5 exons; The *RsSUC* genes in SUT4 all contained 5 exons; the majority of *RsSUC* genes in SUT2 contained 14 exons, of which *RsSUC3b* contained 8 exons. The exon numbers of *RsSUC* genes in SUT2 were more than those in SUT1 and SUT4.

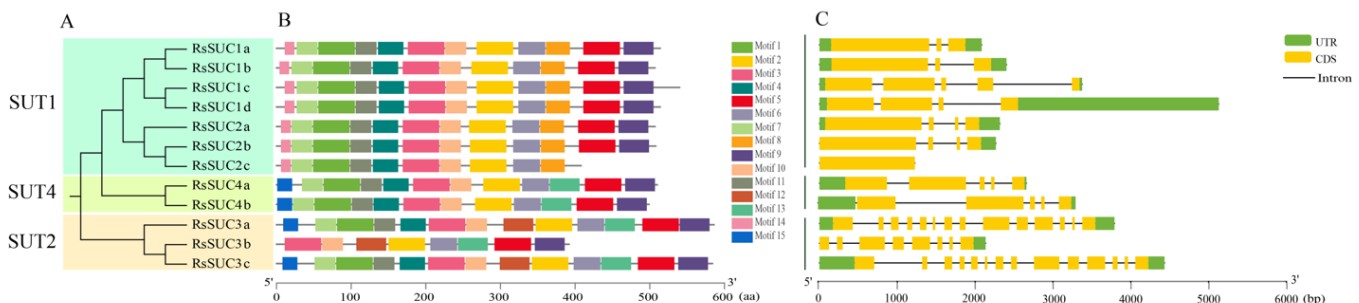

**Figure 2.** Conserved motifs and gene structure analysis of *RsSUC* genes. (**A**) Phylogenetic tree of RsSUC proteins. RsSUC proteins were divided into three subgroups (SUT1, SUT2 and SUT4), and each color group represented a subgroup. (**B**) Conserved motifs of RsSUC proteins. The 15 motifs were identified by using the MEME program, with each number of colored boxes representing a different motif; (**C**) Exon–intron structure of *RsSUC* genes. Boxes and lines represent exons and introns, respectively.

### 3.4. Expression Analysis of RsSUC Genes during Radish Taproot Development

SUT1 is the largest subgroup among *SUC* gene family in radish. To elucidate the functions of SUT1 members in radish, the dynamic expression patterns of SUT1 members in developing leaves and taproots were analyzed by RT-qPCR (Figure 3). The results showed that *RsSUC1a*, *RsSUC1b*, *RsSUC2a*, *RsSUC2b*, and *RsSUC2c* were predominately expressed in leaves, whereas *RsSUC1c* and *RsSUC1d* were expressed in the leaves and taproots. The expression level of *RsSUC1a* and *RsSUC1b* increased significantly at 60 days after sowing (DAS) and decreased significantly at 100 DAS. RsSUC2a, *RsSUC2b*, and *RsSUC2c* were expressed at a higher level in leaves at 20 DAS, whereas the expression level of these genes appeared to be down-regulated afterward. The expression level of *RsSUC1c* and *RsSUC1d* increased significantly in leaves at 80 days after sowing (DAS) and decreased significantly at 100 DAS. In the taproots, *RsSUC1c* was expressed at a lower level at 20 DAS, whereas the expression appeared to be up-regulated afterward. However, the expression level of *RsSUC1d* increased significantly at 40 DAS and 80 DAS, respectively, decrease significantly at 60 DAS and 100 DAS.

In this study, *RsSUC1b* showed a high level of expression during the development of leaves, and it might play a vital role in sucrose transport in leaves. In addition, the nucleotide sequence identity between *RsSUC1b* and *AtSUC1* show as high as 83.40%. Therefore, *RsSUC1b* was selected for further investigation of the function in sugar transport during radish taproot formation.

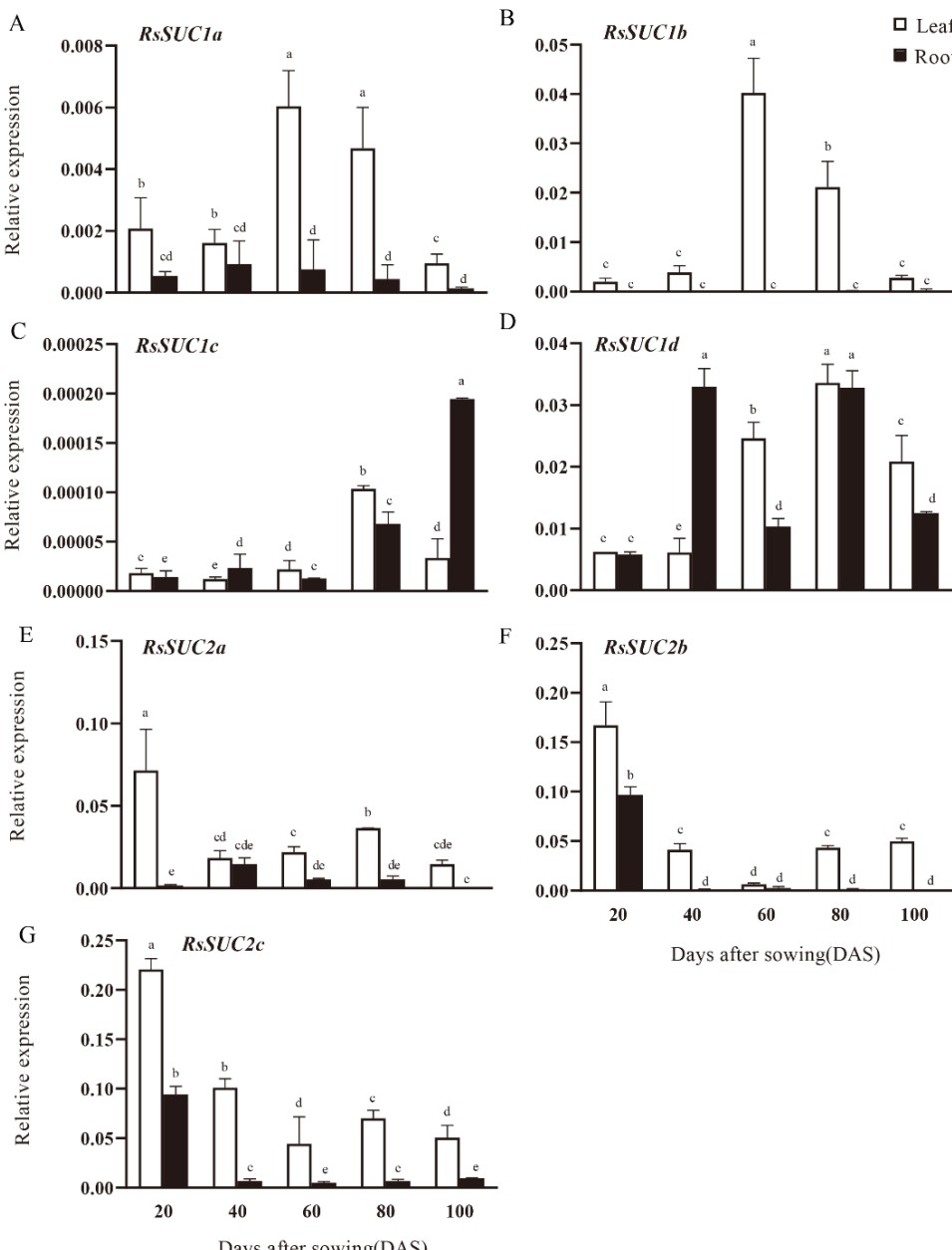

**Figure 3.** Expression patterns of *RsSUC* genes during radish development. (**A–G**) display the relative expression levels of *RsSUC1a*, *RsSUC1b*, *RsSUC1c*, *RsSUC1d*, *RsSUC2a*, *RsSUC2b*, and *RsSUC2c* in leaves and taproots at 20, 40, 60, 80, and 100 days after sowing (DAS), respectively. Each bar shows the mean ± SD (n = 3). Values with different letters indicate a significant difference at $p < 0.05$ according to Duncan's multiple range tests.

*3.5. Subcellular Localization of RsSUC1b*

To confirm the intracellular localization of RsSUC1b protein, *35S::RsSUC1b-GFP* expression vector containing green fluorescent protein (GFP) reporter gene was constructed. The fusion vector *35S::RsSUC1b-GFP* and a plasma membrane marker protein, auxin transporter *35S::AUX1-RFP* fusion protein, co-infiltrated in tobacco leaf epidermal cells by *Agrobacterium* mediated transformation. The results showed that the GFP signals transiently expressed by *35S::RsSUC1b-GFP* overlapped completely with the *35S::AUX1-RFP*, whereas the control displayed GFP signal in the plasma membrane and nucleus of cells, indicating that *RsSUC1b* might function in the plasma membrane (Figure 4).

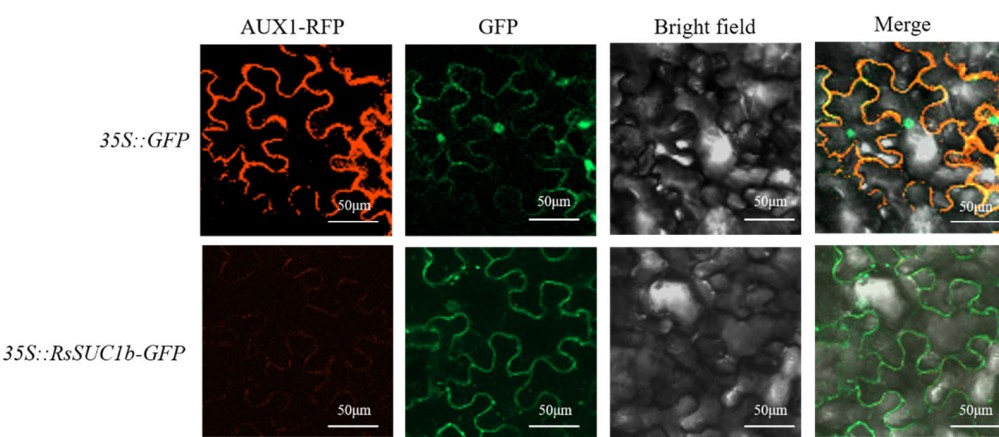

**Figure 4.** The subcellular localization of *RsSUC1b* in tobacco epidermal cells. The plasmids *35S::GFP* and *35S::AUX1-RFP* were used as the negative control and cell membrane marker, respectively. Scale bars = 50 μm.

### 3.6. Effects of RsSUC1b Overexpression on Seedling Growth under Different Sucrose Treatments

The transgenic *Arabidopsis* plants overexpressing *RsSUC1b* were generated to further investigate the biological function of *RsSUC1b*. Homozygous plants were obtained from three generations (T$_3$) of transgenic lines and confirmed higher transcript levels of *RsSUC1b* (Figure S2). To further verify the role of *RsSUC1b* in sucrose transport, the seedling growth of transgenic and wild-type (WT) plants was examined under different sucrose concentration treatments (0%, 1%, 3% and 6%). All transgenic and WT plants could not grow normally in media without sucrose, indicating the importance of sucrose for the growth of plants. Furthermore, the phenotype has no significant difference between transgenic and WT plants were planted in the media without sucrose. Under the conditions of 1%, 3%, and 6% sucrose concentration, transgenic plants grew more vigorously than WT plants, and showed that the larger rosette leaves and the number of lateral roots was increased. The transgenic seedlings may be subject to osmotic stress on the medium with sucrose concentration of 6%, showing smaller rosette leaves and shorter lateral roots compared with 1% and 3% sucrose concentration (Figure 5A,B). Notably, the number of transgenic lateral roots increased in the medium with a relatively high sugar content compared with WT. As compared with WT plants, the relative expression levels of several root development-related genes, *AtWOX4*, *AtKNAT1*, and *AtLBD3*, were increased in the transgenic plants under sucrose treatments of 6% (Figure 5C). These results show that the overexpression of *RsSUC1b* could increase the uptake efficiency of exogenous sucrose.

### 3.7. Overexpression of RsSUC1b in Arabidopsis Resulted in Early Flowering, Increased Height, and Higher SSCs of Plants

The transgenic lines grew faster and flowered earlier than WT plants (Figure 6A). After 45 d of growth, the height of transgenic plants was significantly higher than that of WT plants (Figure 6B and Figure S3). When transgenic lines and WT plants were planted and grown under the same conditions for 45 d, the soluble sugar including sucrose, fructose, glucose, and total sugar contents were analyzed (Figure 6C). Overexpression of *RsSUC1b* significantly increased the soluble sugar contents (SSCs) in the leaves and pods of the transgenic lines. The total sugar contents increased by 23.08% in the leaves and by 88.36% in the pods compared with the WT. In the transgenic lines, the sucrose content was also considerably increased by 30.13% in the leaves and by 92.89% in the pods. The hexose (fructose and glucose) content was considerably increased in transgenic lines. The fructose and glucose content was increased by 6.69% and 12.32% in the leaves, respectively, and by 28.70% and 133.50% in the pods, respectively, indicating that overexpression of *RsSUC1b* in *Arabidopsis* resulted in phenotypic and SSC variations during both vegetative and reproductive growth stages.

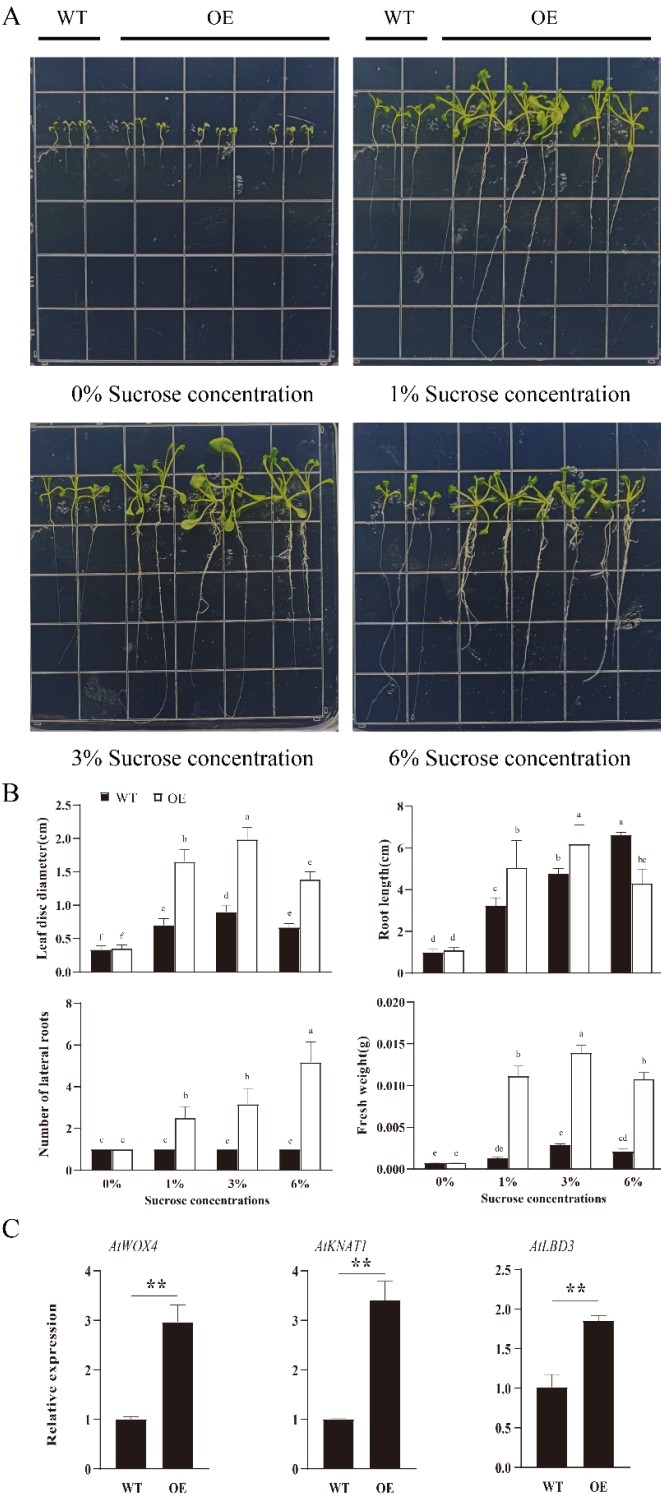

**Figure 5.** *RsSUC1b*-overexpressing (**OE**) *Arabidopsis* plants response to the different sucrose concentrations. The morphology (**A**) and morphological index (**B**) were 20-day-old wide type (WT) and transgenic line seedlings after exogenous sucrose treatment. (**C**) The expression profiles of plant development-related genes of *AtWOX4*, *AtKNAT1*, and *AtLBD3* in WT and transgenic plants under sucrose treatments of 6%. Each bar shows the mean ± SD (n = 3). Values with different letters indicate a significant difference at $p < 0.05$ according to Duncan's multiple range tests. * indicates significant difference at $p < 0.05$, and ** indicates significant difference at $p < 0.01$ according to Student's *t*-test.

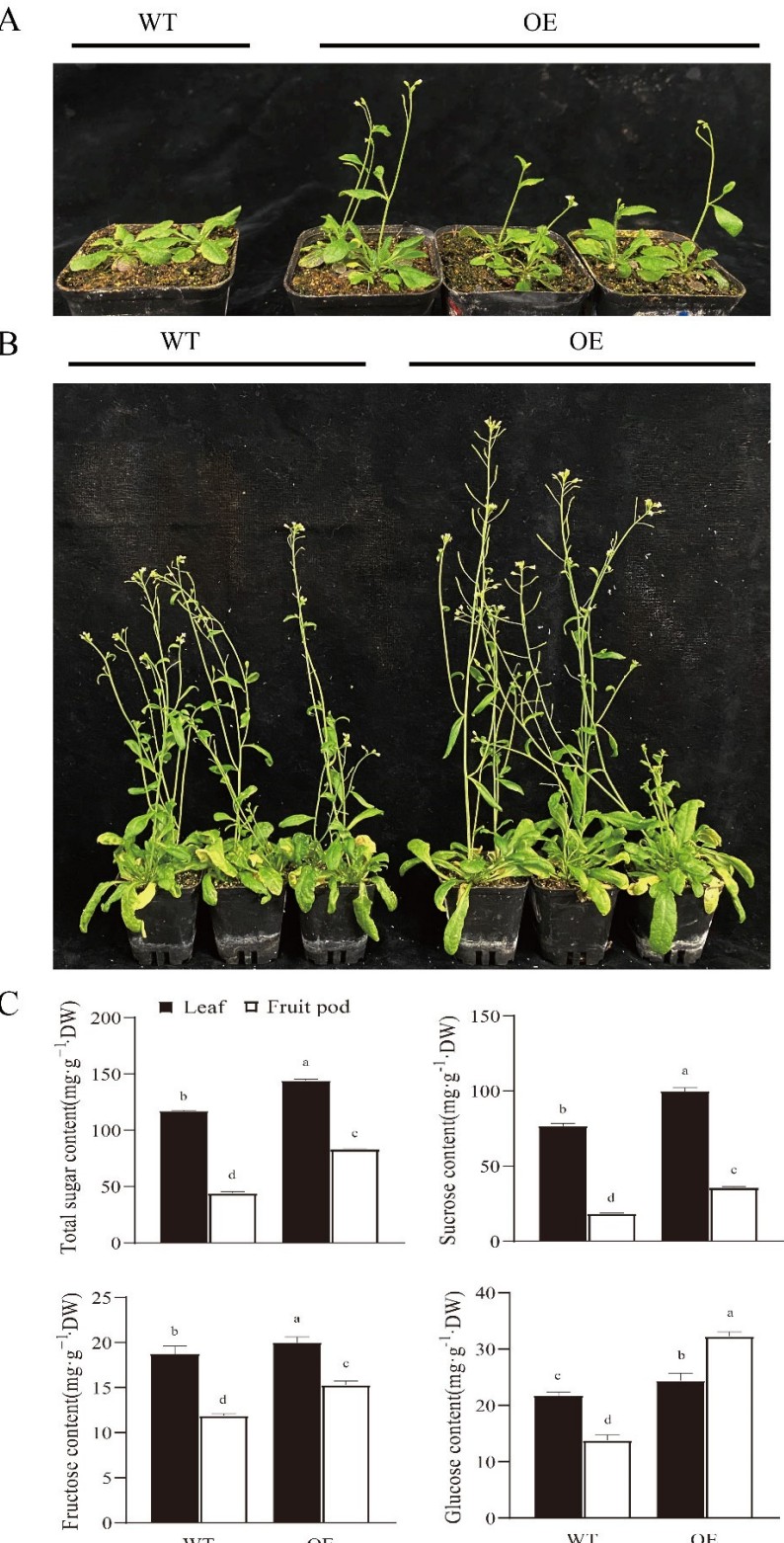

**Figure 6.** The phenotypes of *RsSUC1b*-overexpressing (OE) transgenic *Arabidopsis* plants. (**A**) Morphological phenotypes of WT and three independent transgenic lines in vegetative growth stage (**A**) and in bolting stage (**B**), bar = 1 cm. (**C**) The soluble sugar (total sugar, sucrose, fructose and glucose) contents of the transgenic lines and WT plants in leaves and pods. DW, dry weight. Each bar shows the mean ± SD (n = 3). Value with different letters indicates a significant difference at *p* < 0.05 according to Duncan's multiple range tests.

## 4. Discussion

### 4.1. Characterization of SUC Gene Family Members in Radish

The growth and development of plants depend on the transportation and utilization of sugars produced during photosynthesis. Sucrose transporters (SUTs/SUCs) are a group of membrane proteins that involve in the loading and unloading of sucrose in phloem and play essential roles in plant growth, development and stress response [33,34]. With the completion of genome sequencing, SUC family members have been reported in various plant species. Herein, totally 12 *SUC* genes were identified in radishes based on the genome database in this study (Table S1). The radish genome contains more *RsSUC* genes than the maize [11], sorghum [13], rice [35], and wheat [36] genome, indicating specific expansion events of SUC family may have occurred in radish evolution.

The duplicated genes could obtain new functions to improve plant adaptability to the environment in the process of plant evolution. Phylogenetic analysis showed that *RsSUC* genes were classified into five subgroups. SUT1 subgroup was dicotyledon-specific, and the largest number is in radishes, indicating that it could play a critical role in sugar loading and transportation [17]. The phylogenetic relationship of RsSUCs was also supported by both their conserved motif and gene structure. The RsSUC members in each subgroup shared several unique motifs, indicating that the RsSUC proteins within the same subgroups may have certain functional similarities, and the motif distribution suggested that these genes were largely conserved during evolution [37]. The characteristics of gene structure also had an important influence on the evolution of gene families. The gene structure analysis showed that each subfamily displayed similar exon–intron organizations. Interestingly, SUT1 members contained lower exons, whereas has the longest length, and SUT2 members displayed the opposite result. This might be the result of chromosomal rearrangements and fusions and led to functional diversification of polygenic families [38]. The characterization of the radish *SUC* gene family through genome-wide analysis provided valuable information for further investigating the functions of *RsSUC* genes during radish taproot formation.

### 4.2. Expression Patterns and Functional Diversity of RsSUC Genes

Previous studies revealed that *SUC* genes have distinct expression patterns in various tissues and developmental stages of plants [5]. In this study, SUT1 members showed different expression patterns in leaves and taproots at various points (Figure 3). *RsSUC1a* and *RsSUC1b* showed the highest relative expression levels in leaves at 60 DAS, then declined sharply, indicating that they may play positive roles in regulating the loading of sucrose during leaf development. The weak expression of *RsSUC1c* was observed during leaf and taproot development. *RsSUC1d* was significantly higher in leaves and taproots at 80 DAS than in any other stages, suggesting that it may be involved in the transportation of sucrose during the late developmental stages of leaves and roots. *RsSUC2a*, *RsSUC2b*, and *RsSUC2c* showed higher relative expression levels in leaves at 20 DAS, indicating that they might function on the transportation of sucrose during the early developmental stages of leaves. These results showed that there were differences in expression patterns and functions of *RsSUC* genes, whereas they all participated in the distribution of photosynthetic products and jointly regulated plant growth and development.

### 4.3. Sucrose Transporters Play Fundamental Roles in Plant Growth and Development

Soluble sugar is crucial for plant growth and development, and it usually needs to be transported to sinks [39]. SUCs are involved in transmembrane transport during phloem loading and unloading, for which the function of SUCs in the loading of source organs has been verified in a variety of plants. For instance, overexpressing spinach *SoSUT1* in potatoes promoted the export of sucrose in leaves and significantly increased the contents of soluble sugar in tubers [20]. Inhibiting the expression of potato *StSUT4* resulted in plant growth retardation, a large amount of soluble sugar and starch accumulated in leaves, and finally reduced tuber yield [40]. In this study, *RsSUC1b* showed the highest relative

expression level in leaves than taproots of radishes, and RsSUC1b protein was localized on the plasma membrane. In *Arabidopsis* and other species, SUT1 gene functions in the cell membrane [19,41]. Therefore, it could be speculated that *RsSUC1b* gene might function in the cell membrane and participate in the transmembrane transport of sucrose.

The transgenic plants grew more robustly than WT plants on the medium with sucrose concentrations of 1%, 3%, and 6%. With the increase of sucrose concentration, transgenic plants had more lateral roots than WT. In root crops, the yields of storage roots are mainly determined by secondary growth driven by the vascular cambium. It was reported that *AtWOX4*, *AtKNAT1*, and *AtLBD3* were positive regulators of cambial activities in *Arabidopsis* [42]. In this study, as compared with WT, the lateral root number of transgenic plant was increased in the medium with high sucrose concentration (6%). As compared with WT plants, the relative expression levels of these genes were up-regulated in the transgenic plants. These results demonstrated overexpression of *RsSUC1b* increased the absorption efficiency of exogenous sucrose, and the source organ (leaves) obtained more carbohydrates which could be transported to the sink organ (root). Therefore, it is reasonable to infer that the *RsSUC1b* gene could promote sugar accumulation in the roots of plants, which induces the higher expression level of several root development-related genes such as *AtWOX4*, *AtKNAT1*, and *AtLBD3* in transgenic plants.

In addition, the overexpression of *RsSUC1b* resulted in the increased SSCs in leaves and pods, indicating that overexpression of *RsSUC1b* could influence sucrose metabolism and sugar accumulation. SUCs not only are involved in sucrose transport but also play essential roles in pollen germination, fruit ripening, and reproductive growth in various plant species [35,43]. In this study, overexpression of *RsSUC1b* produced a visible phenotype in the shoot, transgenic *Arabidopsis* plants had the feature of early bolting and flowering. The role of *SUC* gene in flowering has been verified in several species. The overexpression of apple *MdSUT2* gene in *Arabidopsis* led to early flowering and increased plant height [44]. The antisense inhibition of sucrose transporter activity in phloem of tobacco resulted in delayed flowering.

## 5. Conclusions

In conclusion, the systematic genome characterization of *RsSUC* gene family was investigated in this study, and the expression patterns of each member were different during radish growth and development. *RsSUC1b* might be a plasma membrane transporter. In addition, overexpression of *RsSUC1b* affected sucrose transport and sugar accumulation in *Arabidopsis* and promoted vegetative and reproductive growth. These results might lead to a better understanding of the function of *SUC* genes in sugar transportation during radish taproot formation.

**Supplementary Materials:** The following are available online at https://www.mdpi.com/article/10.3390/horticulturae8111058/s1, Table S1: List of the 12 *RsSUC* genes identified in this study; Table S2: List of SUC proteins used for the construction of the phylogenetic tree; Table S3: The primer sequences of RT-qPCR; Table S4: Primer sequences for *35::RsSUC1b-GFP* construction; Figure S1: The phenotype during the development of radish. Figure S2: RT-PCR analysis of WT and over-expression T3 transgenic *Arabidopsis* plants; Figure S3: The plant height of WT and over-expression of *RsSUC1b* *Arabidopsis* plants.

**Author Contributions:** Conceptualization, X.Z. (Xiaofeng Zhu), X.Z. (Xiaoli Zhang) and L.L.; writing—original draft preparation, X.Z. (Xiaofeng Zhu) and Y.C.; writing—review and editing, Y.M., L.W., L.X., Y.W. and L.L.; visualization, X.Z. (Xiaofeng Zhu), X.Z. (Xiaoli Zhang) and R.X.; validation, X.Z. (Xiaofeng Zhu) and X.Z. (Xiaoli Zhang); project administration, L.L. and R.L. All authors have read and agreed to the published version of the manuscript.

**Funding:** This work was supported by the Jiangsu Agricultural S&T Innovation Fund [CX(21) 2020], the "JBGS" Project of Seed Industry Revitalization in Jiangsu Province (JBGS(2021)071), and the earmarked fund for Jiangsu Agricultural Industry Technology System [JATS(2022)463], the Guidance Foundation, the Hainan Institute of Nanjing Agricultural University(NAUSY-MS02), the Project Founded by the Priority Academic Program Development of Jiangsu Higher Education Institutions (PAPD).

**Institutional Review Board Statement:** Not applicable.

**Informed Consent Statement:** Not applicable.

**Data Availability Statement:** Not applicable.

**Conflicts of Interest:** The authors declare that they have no known competing financial interests or personal relationships that could have appeared to influence the work reported in this paper.

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
