# Peer review of "Genome-Wide Identification of Sucrose Transporter Genes and Functional Analysis of RsSUC1b in Radish (Raphanus sativus L.)"

_horticulturae, doi:10.3390/horticulturae8111058_

Round 1

Reviewer 1 Report

The authors present a nice series of experiments focused on sucrose transport genes in radish that I think would be very interesting to readers of Horticulturae. However, the authors need to do a lot of work on the organization and explanation of their results as presented in this article. The authors have a great hook in why to study sucrose transport in radish (e.g., its taproot and how that affects its growth and distribution of sugars), but they do not bring this out. The authors also have several instances in the results section that need to be moved to the discussion section, and a very dry discussion section that instead reads as a results section. With only this paper to go by, it is unclear to me what the authors think are the most exciting takeaways for the reader. Most problematic for me are the presentation of the results. Sequences of the radish genes analyzed should be included in the supplement and perhaps even uploaded onto NCBI – it is clunky to try and find them in the online radish genome database, and as the genes are the centerpiece of this paper, their sequences need to be as readily accessible as possible. Figure captions are extremely lacking, and there is also no description of what statistical tests were actually used, only that there was significance and that certain software were used. This paper cannot be accepted the way it is, though I think authors should be able to clean it up and address these concerns, allowing for resubmission. My more specific comments are below.

More specific comments

I suggest leading the paper with the third paragraph of the introduction (e.g., making it the first paragraph instead), because sucrose transport is especially interesting in radish due to its unique taproot, so that better motivates the study than just jumping in and defining sucrose and sucrose transporters more abstractly, with nothing to make readers interested about that topic.

In the methods section, authors state that their discovered sequences were “submitted to NCBI” but I have found no accession numbers to show this, and perhaps the authors mean something else? Did they instead BLAST their sequences against all sequences on NCBI? But furthermore, in the journal policies of Horticulturae, it seems that it is a requirement that sequences are submitted to NCBI or equivalent database at some point so accession numbers can be included in the finished article, and I do not think the authors have done this. (if so, they should indicate this in their data availability statement, which currently contains “not applicable”)

In section 2.4 of the methods section, it is unclear to me what “homogenization of the internal reference gene” means – there should certainly be a reference gene used to standardize interpretation of the expression of the gene of interest, but I do not know what “homogenization” means in the context of this and authors should give more detail.

The first part of section 2.7 needs some editing and more explanation – there are not full sentences and I cannot tell what was being done to extract the sugar contents. For example – dried sample of what? The whole radish? Parts of it? How was it dried?

Section 2.8, on statistical analysis, should indicate which statistical tests were completed using which software, rather than lumping everything together with no indication of specific tests.

In section 3.1, this sentence stood out to me: “Some proteins with few transmembrane regions might have not evolved completely or be partially degenerated”. I think the authors are conflating biological reasons (“might not have evolved completely”) and technical issues (“be partially degenerated”, which I am assuming means perhaps the gene wasn’t captured fully in sequencing?). Authors need to separate out these reasons, and do it in the discussion rather than the results section.

For figure 1, I suggest bolding the names of the 12 Rs sequences – it’s very difficult to pick them out with the symbols alone. The color catches the eye better than the symbols.

For all figures throughout the paper – all figures throughout the paper actually need an informative, stand alone caption that describes each panel of the figure. Currently it is as if the figure captions only have a title and nothing further. For figure 1: colors need to be named in the caption. For figure 2: Some indication of what colors mean and how each figure was generated. For figure 3: each graph needs its own panel letter, and each needs a description in the caption. For figure 4: each image needs its own panel letter and needs a description in the caption, as well as description of what microscope and what magnification was used to capture the image.

For figure 2: The tree in the first panel needs to be remade at higher resolution, and bootstrap values should also be included on the branches since the methods describe that bootstrapping was done.

Section 3.3 also contains statements that would be better placed in the discussion, such as this one: “suggesting that these proteins may have similar functions and further supported the group classification”. A statement like this also needs a citation – I am not convinced that similar motifs always mean similar functions.

In Figure 4: The caption is insufficient so I have no idea what I am looking at. The text in the corresponding section suggests that only the control contains GFP signal in the nucleus, but in the GFP channel on the bottom row from the construct, it looks as if there IS a nucleus-shaped object in bright green in the upper right corner. A DAPI stain may be more convincing to show me what is and is not a nucleus.

Again in section 3.6, there is a statement better suited to the discussion section: “This results show that the overexpression of RsSUC1b could increase the uptake efficiency of exogenous sucrose and some genes related to lateral roots development might be induced by high sucrose content in transgenic plants, further confirmed though RT-qPCR (Figure 5C).”

Again in Figures 5 and 6, as I’ve said earlier, the figure caption needs to be more descriptive. Authors needs to define the abbreviation “OE” which is in panel A. In Panels B and C, each graph needs to be explained explicitly in the caption. Also, rather than simply indicating statistical significance, the statistical test used needs to be indicated somewhere. Sample sizes also need to be indicated for all graphs.

The first paragraph of the discussion should not so closely echo your introduction – it needs to give more a summary of what you found in radish rather than again setting up the pitch for finding things out in radish.

The entire discussion section 4.1 reads as a rehash of the results, rather than interpreting your results and placing them in larger context, authors are just re-stating each of their specific findings.

The discussion as a whole needs a lot of work to read as a discussion and show why this study did some awesome things. Currently it is too heavy on the results and with not enough emphasis on what this all means.

I think the authors need to emphasize the fact that they even functionally examined what some of their SUC genes from radish did in various expression vectors – they went a step farther than simply identifying the genes and that deserves to be highlighted. This is related to my suggestion for the introduction – bring out the fact that they’re studying radish with its taproot system and what that unique adaptation (which tobacco and Arabidopsis, for instance) might do differently with the SUC genes, and then indeed there are differences when those genes are expressed in those other vectors. Really think critically about the interesting things about the findings for the discussion, rather than a straight repetition of every single one of the results, in seemingly the same order as we just saw in the results section.

Author Response

Dear editor,
Many thanks for the editor’s and reviewers’ positive and constructive comments on this manuscript (horticulturae-1938283) entitled with ‘Genome-Wide Identification of Sucrose Transporter Genes and Functional Analysis of RsSUC1b in Radish (Raphanus sativus L.)’
Based on the suggestions and comments, we revised the manuscript carefully and thoroughly, and made corresponding corrections in the revised version of this manuscript. We have addressed the revisions and corrections in the followed Point-by-Point Authors’ Response to Comments.

Reviewer 2 Report

I have main concerns and questions regarding the rationale of choosing RsSUC1b and the inadequate description of methods in this study.

In Figure 1, RsSUC1a, b, c d and AtSUC1 are clustered in the same group. What are the sequence similarities among RsSUC1b, c, d and AtSUC1? In figure 2, it seems that RsSUC1a and RsSUC1b share the highest similary in term of motif and gene structure. Do they differ at nucleotide sequence level? If not, how did you design qRT-PCR primers for specific RsSUC1b amplification?

What is the life cycle of your radish growth? What is the developmental pattern of radish leaves? Without those information, it is hard to comprehend the difference between samples from 60 DAS and those from 100 DAS. More information should be provided to support your choice of RsSUC1b for gene functional characterization.

What is the expression pattern (temporal and spatial) of AtSUC1? The reference for the sentence "It was found that the AtSUC1 act as invital (?) role by transporting the sucrose through transmembrane in the pollen tube and root tip" is missing.

Section 4.1

"...indicating specific amplication (?) events..." change to " ...indicating specific expansion events..."

Figure 5C  What are those genes? Which tissue is used for quantifying their expression? Just mentioning them as root development-related is too vague. Please briefly describe them and provide references accordingly. In addition, RsSUC1b is predominantly expressed in leaves. How does its expression affect those root-related genes?

Author Response

Point-by-Point Response

Manuscript ID: horticulturae-1938283

Title: Genome-Wide Identification of Sucrose Transporter Genes and Functional

Analysis of RsSUC1b in Radish (Raphanus sativus L.)

Authors: Xiaofeng Zhu, Xiaoli Zhang, Yang Cao, Ruixian Xin, Yinbo Ma, Lun

Wang, Liang Xu, Yan Wang, Rui Liu, Liwang Liu *

Received: 12 September 2022

Dear editor,

Many thanks for the editor’s and reviewers’ positive and constructive comments on this manuscript (horticulturae-1938283) entitled with ‘Genome-Wide Identification of Sucrose Transporter Genes and Functional Analysis of RsSUC1b in Radish (Raphanus sativus L.)’

Based on the suggestions and comments, we revised the manuscript carefully and thoroughly, and made corresponding corrections in the revised version of this manuscript. We have addressed the revisions and corrections in the followed Point-by-Point Authors’ Response to Comments.

Response to Reviewer 2 Comments:

Some punctual requests are:

Point 1: In Figure 1, RsSUC1a, b, c d and AtSUC1 are clustered in the same group. What are the sequence similarities among RsSUC1b, c, d and AtSUC1? In figure 2, it seems that RsSUC1a and RsSUC1b share the highest similary in term of motif and gene structure. Do they differ at nucleotide sequence level? If not, how did you design qRT-PCR primers for specific RsSUC1b amplification?

Response 1: Many thanks for the reviewer’s positive comments. RsSUC1a, RsSUC1b, RsSUC1c, RsSUC1d and AtSUC1 share the 85.83, 85.05, 80.07 and 83.91% identity at protein level, respectively. RsSUC1a and RsSUC1b share the 87.90% identity at nucleotide sequence level. NCBI website was employed to design the RT-qPCR primers of RsSUC genes, and detected the specificity.

Gene name

Gene ID

Forward primer (5'-3')

Reverse primer (5'-3')

RsSUC1a

Rs099020

CTCCTCGTCACCATCACGTC

AAGAAACGGGAACCATGCGA

RsSUC1b

Rs455130

CCGCTGGAGACGCTAAAAGA

ATCGCAAGCGTTTGTCATCG

Point 2: What is the life cycle of your radish growth? What is the developmental pattern of radish leaves? Without those information, it is hard to comprehend the difference between samples from 60 DAS and those from 100 DAS. More information should be provided to support your choice of RsSUC1b for gene functional characterization.

Response 2: Many thanks for the reviewer’s constructive suggestions. We added the sample information of radish in section 2.1 according to reviewer’s suggestions. (Page 2, lines 34).

“The phenotypes during the development stages of radish were shown in Figure S1. The growth cycle of radish is about 3 months. According to the phenotypic changes at each stage, the development stages of radish could be divided into cortex split stage (20-40 DAS), early stage of taproot thickening (40-60 DAS), middle stage of taproot thickening (60-80 DAS) and late stages of taproot thickening (80-100 DAS). About 0.3 g sample was rapidly frozen in liquid nitrogen and stored at -80℃ for gene expression analysis, the remaining sample was dried at 80℃ for soluble sugar contents determination.”

In this study, RsSUC1b showed a high level of expression during the development of leaves, and it might play a very important role in sucrose transport in leaves. In addition, the nucleotide sequence identity between RsSUC1b and AtSUC1 show as high as 83.40%. Arabidopsis sucrose transporter AtSUC1 was founded to function in sucrose transport during development of pollen and roots (Stadler et al., 1999; Stadler et al., 2008). Therefore, RsSUC1b was selected for further investigation of the function in sugar transport during radish taproot formation.

Figure S1. The phenotype during the development of radish. 20 d, 40 d, 60 d, 80 d and 100 d represents 20, 40, 60, 80 and 100 days after sowing, respectively (20, 40, 60, 80 and 100 DAS).

Reference

Stadler, R.; Truernit, E.; Gahrtz, M.; Sauer, N. The AtSUC1 sucrose carrier may represent the osmotic driving force for anther dehiscence and pollen tube growth in Arabidopsis. The Plant Journal. 1999, 19, 269-278.

Sivitz, A.B.; Reinders, A.; Ward, J.M. Arabidopsis sucrose transporter AtSUC1 is important for pollen germination and sucrose-induced anthocyanin accumulation. Plant Physiol. 2008, 147, 92-100.

Point 3: What is the expression pattern (temporal and spatial) of AtSUC1? The reference for the sentence "It was found that the AtSUC1 act as invital (?) role by transporting the sucrose through transmembrane in the pollen tube and root tip" is missing.

Response 3: Many thanks for the reviewer’s constructive suggestions. AtSUC1 was expressed in pollen, and AtSUC1 has been proposed to function in sucrose uptake during germination in pollen. Furthermore, AtSUC1 is also expressed in roots, and exogenous application of sucrose increased AtSUC1 expression in roots (Stadler et al., 1999; Sivitz et al., 2008). In this revised manuscript, we revised the corresponding sentence in section 3.4 as follow. (Page 7, lines 35).

“It was found that the AtSUC1 act as invital role by transporting the sucrose through transmembrane in the pollen tube and root tip. The nucleotide sequence identity between RsSUC1b and AtSUC1 show as high as 83.40%. In this study, RsSUC1b was expressed at a higher level during the development of leaves, and it may play a very important role in sucrose transport in leaves. was revised as In this study, RsSUC1b showed a high level of expression during the development of leaves, and it might play a vital role in sucrose transport in leaves. In addition, the nucleotide sequence identity between RsSUC1b and AtSUC1 show as high as 83.40%.

Reference

Stadler, R.; Truernit, E.; Gahrtz, M.; Sauer, N. The AtSUC1 sucrose carrier may represent the osmotic driving force for anther dehiscence and pollen tube growth in Arabidopsis. The Plant Journal. 1999, 19, 269-278.

Sivitz, A.B.; Reinders, A.; Ward, J.M. Arabidopsis sucrose transporter AtSUC1 is important for pollen germination and sucrose-induced anthocyanin accumulation. Plant Physiol. 2008, 147, 92-100.

Point 4: Section 4.1 "...indicating specific amplication (?) events..." change to " ...indicating specific expansion events..."

Response 4: Many thanks for the reviewer’s constructive suggestions. In this revised manuscript, we revised the corresponding sentence in section 4.1 as follow. (Page 12, lines 24).

“The radish genome contains more RsSUC genes than the Arabidopsis, rice, sorghum, maize and wheat genome, indicating specific amplification events of SUC family may have occurred in radish evolution [32-33].” was revised as “The radish genome contains more RsSUC genes than the maize [11], sorghum [13], rice [35] and wheat [36] genome, indicating specific expansion events of SUC family may have occurred in radish evolution.

Point 5: Figure 5C What are those genes? Which tissue is used for quantifying their expression? Just mentioning them as root development-related is too vague. Please briefly describe them and provide references accordingly. In addition, RsSUC1b is predominantly expressed in leaves. How does its expression affect those root-related genes?

Response 5: Many thanks for the reviewer’s constructive suggestions. In root crops, the yields of storage roots are mainly determined by secondary growth driven by the vascular cambium. In relation to this, a dynamic yet intricate gene regulatory network (GRNs) should operate in the vascular cambium, in coordination with environmental changes. Common GRNs involved in the vascular cambium-driven secondary growth in storage roots was discussed using the wealth of information available in Arabidopsis (Hoang et al., 2020). It was reported that AtWOX4, AtKNAT1 and AtLBD3 were positive regulators of cambial activities. In this study, the transgenic and wild-type (WT) plants under sucrose treatments of 6%, were used for quantifying these genes expression.

RsSUC1b is a plasma membrane transporter and participates in the transmembrane transport of sucrose. Overexpression of RsSUC1b increased the absorption efficiency of exogenous sucrose, and more carbohydrates were obtained from the source organ (leaves) and transported to the sink organ (root). Therefore, the expression of these root development-related genes might be induced.”

Accordingly, we added the following sentence to section 4.3. (Page 16, lines 25)

“In root crops, the yields of storage roots are mainly determined by secondary growth driven by the vascular cambium. It was reported that AtWOX4, AtKNAT1 and AtLBD3 were positive regulators of cambial activities in Arabidopsis [42]. In this study, as compared with WT, the lateral root number of transgenic plant was increased in the medium with high sucrose concentration (6%). As compared with WT plants, the relative expression levels of these genes were up-regulated in the transgenic plants. These results demonstrated overexpression of RsSUC1b increased the absorption efficiency of exogenous sucrose, and the source organ (leaves) obtained more carbohydrates which could be transported to the sink organ (root). Therefore, it’s reasonably to infer that the RsSUC1b gene could promote source accumulation in the roots of plants, which induced the higher expression level of several root development-related genes such as AtWOX4, AtKNAT1 and AtLBD3 in transgenic plants.”

Reference

Hoang, N.V.; Park, C.; Kamran, M.; Lee, J.Y. Gene regulatory network guided investigations and engineering of storage root development in root crops. Front Plant Sci. 2020, 17, 762.

Yu, C.; Sun, C.; Shen, C.; Wang, S.; Liu, F.; Liu, Y.; Chen, Y.; Li, C.; Qian, Q.; Aryal, B.; Geisler, M.; Jiang, de. A.; Qi, Y. The auxin transporter, OsAUX1, is involved in primary root and root hair elongation and in Cd stress responses in rice (Oryza sativa L.). Plant J. 2015, 83, 818-30.

Stadler, R.; Sauer, N. The Arabidopsis thaliana AtSUC2 gene is specifically expressed in companion cells. Botanica Acta, 1996, 109, 299-306.

Kühn, C.; Hajirezaei, M.R.; Fernie, A.R.; Roessner-Tunali, U.; Czechowski, T.; Hirner, B.; Frommer, W.B. The sucrose transporter StSUT1 localizes to sieve elements in potato tuber phloem and influences tuber physiology and development. Plant Physiol., 2003, 131, 102-113.

Round 2

Reviewer 2 Report

The authors have adequately addressed all my concerns and comments in the revised version of this manuscript.